# Effects of *ESA_00986* Gene on Adhesion/Invasion and Virulence of *Cronobacter sakazakii* and Its Molecular Mechanism

**DOI:** 10.3390/foods12132572

**Published:** 2023-06-30

**Authors:** Yufei Fan, Ping Li, Dongdong Zhu, Chumin Zhao, Jingbo Jiao, Xuemeng Ji, Xinjun Du

**Affiliations:** 1State Key Laboratory of Food Nutrition and Safety, College of Food Science and Engineering, Tianjin University of Science and Technology, Tianjin 300457, China; 17908002@mail.tust.edu.cn (Y.F.); zoelxx@126.com (P.L.); duiduiz@163.com (D.Z.); zcm3278@163.com (C.Z.); 18753391899@163.com (J.J.); 2Tianjin Key Laboratory of Food Science and Health, School of Medicine, Nankai University, Tianjin 300071, China; jixuemeng@nankai.edu.cn

**Keywords:** *Cronobacter sakazakii*, *ESA_00986*, adhesion/invasion, virulence

## Abstract

*Cronobacter sakazakii* is an opportunistic Gram-negative pathogen that has been identified as a causative agent of severe foodborne infections with a higher risk of mortality in neonates, premature infants, the elderly, and immunocompromised populations. The specific pathogenesis mechanisms of *C. sakazakii*, such as adhesion and colonization, remain unclear. Previously, we conducted comparative proteomic studies on the two strains with the stronger and weaker infection ability, respectively, and found an interesting protein, ESA_00986, which was more highly expressed in the strain with the stronger ability. This unknown protein, predicted to be a type of invasitin related to invasion, may be a critical factor contributing to its virulence. This study aimed to elucidate the precise roles of the *ESA_00986* gene in *C. sakazakii* by generating gene knockout mutants and complementary strains. The mutant and complementary strains were assessed for their biofilm formation, mobility, cell adhesion and invasion, and virulence in a rat model. Compared with the wild-type strain, the mutant strain exhibited a decrease in motility, whereas the complementary strain showed comparable motility to the wild-type. The biofilm-forming ability of the mutant was weakened, and the mutant also exhibited attenuated adhesion to/invasion of intestinal epithelial cells (HCT-8, HICE-6) and virulence in a rat model. This indicated that *ESA_00986* plays a positive role in adhesion/invasion and virulence. This study proves that the *ESA_00986* gene encodes a novel virulence factor and advances our understanding of the pathogenic mechanism of *C. sakazakii*.

## 1. Introduction

*Cronobacter* spp. is a motile Gram-negative bacillus with the ability to form biofilm. It is a facultative anaerobic bacterium [1]. Among the seven species of this genus, *Cronobacter sakazakii* exhibits the highest prevalence and clinical relevance in the human population [2,3]. It is considered a causative agent of neonatal meningitis and necrotizing colitis [4], which can result in a high mortality rate of up to 80% [5,6]. In addition to its impact on neonates, *C. sakazakii* has the ability to induce severe infections in elderly individuals and people with compromised immune systems [7]. Numerous studies have suggested that *C. sakazakii* is often present in different food types and most clinical cases are associated with powdered infant formula (PIF) and other infant food products [8].

In recent years, significant advancements have been made in the study of the pathogenic mechanism of *C. sakazakii*. Numerous studies have provided evidence that *C. sakazakii* is capable of adhering to and invading human intestinal epithelial cells. Additionally, it has the ability to cross the blood–brain barrier and replicate within macrophages [9,10]. As a peroral pathogen that can cause systemic infections, *C. sakazakii* must have the necessary virulence factors to invade various epithelial and endothelial cells in human and animal hosts, evade host defense mechanisms, and traverse the blood–brain barrier. Several virulence determinants have been identified and validated, such as lipopolysaccharides (LPSs), outer membrane protein A (Omp A), and outer membrane protein X [11]. LPSs play a crucial role in the invasion of intestinal epithelial cells by *C. sakazakii* via the disruption of tight junctions [12,13,14]. Omp A can interact with glycoproteins on the surface of human brain microvascular endothelial cells (HBMECs) to mediate adhesion infection [15,16]. Omp X has been confirmed to play a critical role in adhesion to and invasion of Caco-2 cells [17,18]. The flagella of *C. sakazakii* serve as immune stimuli and can trigger the production of pro-inflammatory cytokines in human-derived monocytes [19]. LysR plays a critical role in the regulation of factors involved in adhesion to and invasion of human intestinal cells [20]. However, the details of the adhesion and infection mechanisms of *C. sakazakii* are still unclear.

Adhesins are one of the most important factors in bacterial adhesion. Some adhesins exist on fimbriae, which are conducive to the initial adhesion of bacteria to cells. Some adhesins can recognize and interact with receptors on the cell surface, triggering a series of signals that rearrange the cytoskeleton of the host cell and induce bacterial uptake. The Intimin/Invasin (Int/Inv) family encompasses a wide range of proteins that facilitate bacterial attachment to and/or invasion of host cells. These proteins play a crucial role in mediating the interaction between the bacterium and its target cells [21]. The Int/Inv family’s prototype members originate from pathogenic strains of *Escherichia coli* (Int) [22] and *Yersinia* (Inv) [23]. These strains have been extensively studied and serve as important models for understanding the mechanisms of bacterial attachment and invasion in various host cell types. Jerse et al. first describe intimin in enteropathogenic and *E. coli* strains in [24]; it facilitates bacterial entry into eukaryotic cells by engaging in high-affinity binding with members of the β1 integrin family [25]. Some intimin/invasin proteins contain multiple bacterial Ig-like domains that are classified as belonging to the Big_1 superfamily [26]. Ig-like domains are present in the surface proteins of bacteria and have been implicated in bacterial pathogenesis [27]. Previous studies have reported that *Yersinia* or *Salmonella* invasins, which possess Ig domains, are involved in the invasion of different epithelial host cells such as M and Hep-2 cells [28,29]. 

In our previous study, we conducted comparative proteomic studies on the two strains with the stronger and weaker infection ability, respectively. We found an adherence-related protein, ESA_00986, which was highly expressed in the strain with the stronger ability. It has typical Ig-like domains typical of intimin/invasin and is presumed to be a protein in the intimin/invasin family. However, the homolog which is most similar to protein ESA_00986 is the intimin in *E. coli* [24], which is only 32% similar. Therefore, the detailed roles of this gene were unknown. However, the detailed roles of this gene have not been studied. In this study, the functions of the *ESA_00986* gene in *C. sakazakii* were explored via gene knockout and complementation. This study plays a significant part in understanding the detailed role of the ESA_00986 gene in the interactions between *C. sakazakii* and host cells, which is greatly helpful in controlling the foodborne pathogen.

## 2. Experimental Materials and Procedures

### 2.1. Strains and Plasmids

The strains and plasmids employed in this investigation are presented in Table 1. Bacteria were stored in LB broth containing 15% glycerol at −80 °C and grown on LB agar medium and in LB broth at 37 °C while being continuously shaken at 200 rpm. The broth contained chloramphenicol and ampicillin at a concentration of 100 μg/mL each.

### 2.2. Construction of ESA_00986 Mutant

The generation of a mutant with a deletion in the *ESA_00986* gene was carried out using a method previously established in [30]. To linearize pCVD442, PCR was performed using primers pCVD442F and pCVD442R (Table 2). The up and down homologous arms of *ESA_00986* were amplified via PCR using the two pairs of primers (*ESA_00986* QF/*ESA_00986* QR and *ESA_00986* HF/*ESA_00986* HR) listed in Table 2. To construct the pCVD442-QH vector, the fragments located upstream and downstream of the target region were inserted into the linearized pCVD442 suicide vector using a seamless cloning kit (VAZYME, China). Subsequently, the constructed vector was chemically transformed into *E. coli* S17 cells. In the next stage, the vector containing the targeted fragments was transformed into the wild-type (WT) strain of *C. sakazakii* ATCC BAA-894 via chemical transformation. All primer sequences used in these procedures can be found in Table 2.

### 2.3. Complementation

The low-copy vector pACYC184 was used to complement the deletion mutants. To construct the complementary plasmid pACYC184-*ESA_00986*, the primers *ESA_00986* 1F and *ESA_00986* 1R (Table 2) were employed. To construct the *ΔESA_00986* complementation, the pACYC184 plasmid was used, which carried an *ESA_00986* gene fragment. The plasmid was introduced into the *ΔESA_00986* mutant strain, resulting in the complementary strain *cpESA_00986* harboring the *ESA_00986* gene.

### 2.4. Growth Curve Analysis

The bacterial growth curve was determined using established protocols as described previously [31]. In brief, the WT strain, *ΔESA_00986*, and *cpESA_00986* were cultured overnight in LB medium at 37 °C, whereupon the cultures were transferred into 100 mL of fresh culture medium (1:100). Bacterial growth was monitored by measuring the optical density (OD) at 600 nm at hourly intervals for a total of 14 h using a UV-Vis spectrophotometer. Subsequently, the collected data points were used to plot the bacterial growth curves. Each sample was replicated three times to ensure the accuracy and reliability of the obtained results.

### 2.5. Motility Analysis

The motility of bacteria was evaluated by measuring the migration radius in LB containing 0.3% agar [32]. The WT, ΔESA_00986, and the cpESA_00986 strain were incubated at 37 °C overnight for cultivation. Subsequently, the bacterial cultures were transferred onto soft agar plates (LB medium containing 0.3% agar) and incubated at 30 °C for 16 h. Afterwards, colony size was observed, and the average migration radius was calculated.

### 2.6. Hydrophobicity Analysis

To assess hydrophobicity, a xylene contact angle assay was performed following previously established methods [33]. Bacterial cells were cultured in LB broth overnight and then subcultured in fresh LB medium at a 1% inoculum. The cultures were grown until they reached an optical density of 600 nm (OD600) within a range from 0.6–0.8, indicating mid-log phase. Next, the bacteria were collected via centrifugation and subsequently washed with phosphate-buffered saline (PBS), then adjusted to OD600 of 0.5. 2 mL bacterial suspension with a xylene mixture, trained for 3 h at room temperature, and then the OD600 of the bacterial cultures was measured using a spectrophotometer. The hydrophobicity of the bacterial cells was calculated using the following formula: [(H0 − H)/H0] × 100%, where H0 and H represent the optical density at 600 nm of the bacterial suspension before and after the addition of xylene.

### 2.7. Outer Membrane Permeability Analysis

Permeability was assessed via N-Phenyl-1-naphthylamine (NPN) assay. Briefly, bacterial cells were initially cultured in LB broth overnight. The culture was then transferred to fresh LB medium at a 1% inoculum and incubated until OD600 reached a range of 0.6-0.8. Cells were harvested and suspended in PBS, then adjusted to OD600 of 0.5. A final concentration of 1 µM of NPN was added to the sample. The fluorescence of the sample was then continuously measured using an excitation wavelength of 380 nm and an emission wavelength of 430 nm.

### 2.8. Biofilm Formation Ability Analysis

Biofilm formation capacity was assessed via crystal violet (CV) staining. In summary, suspensions of the WT, ΔESA_00986, and cpESA_00986 strains were prepared at an OD600 value of 0.7. These suspensions were then added to individual wells and incubated at 37 °C for a duration of 48 h. Following incubation, the biofilm was fixed with methanol after 15 min, and the supernatant was removed. The samples were air-dried at 25 °C. Subsequently, each well was treated with 1% CV solution and incubated for 30 min. After staining, the wells were washed three times, and 200 μL ethanol was added for decolorization purposes. The optical density was measured at a wavelength of 570 nm using a spectrophotometer. Each experiment was performed with three repetitions to ensure the accuracy and reliability of the results.

### 2.9. Adhesion/Invasion Capacity Analysis

The adhesion/invasion assay was conducted following the previously described method with slight modifications [34]. HCT-8 cells and HIEC-6 cells were used for this assay. The cell monolayer was cultured using RPMI-1640 medium and 10% fetal bovine serum. The WT, ΔESA_00986 and cpESA_00986 cells were collected at an OD600 of 0.6, then were washed and suspended in RPMI-1640 medium. Next, 1 mL of *C. sakazakii* bacteria was added to the HIEC-6 cells or HCT-8 cells at a Multiplicity of Infection (MOI) of 100. After incubation for 3 h, the cells were washed with PBS and subsequently lysed using 1% Triton X-100. The resulting cell suspensions were then serially diluted with PBS and plated on LB agar plates for colony counting analysis. 

The invasion assay was conducted following the previously described method with slight modifications [35]. A 1 mL quantity of *C. sakazakii* bacteria was added to the HIEC-6 cells or HCT-8 cells at a Multiplicity of Infection (MOI) of 100. After incubation for 1 h, the cells were washed with PBS and covered with 2 mL RPMI-1640 medium containing gentamicin at bactericidal concentration of 100 µg/mL to kill extracellular bacteria. After 2 h, the cells were washed with PBS and subsequently lysed using 1% Triton X-100. The resulting cell suspensions were then serially diluted with PBS and plated onto LB agar plates for colony-counting analysis.

### 2.10. Rat Virulence Assay

Neonatal rats (48 h old) were sourced from SPF (Beijing) Biotechnology Co., Ltd. The rats were housed in separate clean cages that had been disinfected prior to use. A total of 12 neonatal rats were assigned to each experimental group. To administer the bacteria, each pup received an oral dose of 0.2 mL of infant formula with sheep’s milk powder containing 1 × 109 bacterial cells. This was done using an animal gavage needle. The control group received an oral dose of 0.2 mL of sterile infant formula with sheep’s milk powder only. During the post-challenge period, the pups were closely monitored for physiological symptoms and mortality rates every 2 h. After 48 h, all the pups were euthanized for further analysis. Blood samples were collected and then subjected to centrifugation at 2000× *g* for 15 min at 4 °C. This process allowed for the separation of serum, which was subsequently stored at −80 °C for future analysis. Next, liver, spleen, and brain samples were excised, washed with PBS, and dried with filter papers during necropsy. All tissues were rapidly frozen via immersion in liquid nitrogen and then stored at −80 °C for future utilization. Animal experiments were performed according to the guidelines of the institutional animal ethics committee and supported by the Institutional Animal Committee of Tianjin University of Science and Technology (2022117).

### 2.11. Quantification of Bacteria in Tissues

The tissues intended for *C. sakazakii* enumeration were washed three times with PBS (pH 7.4). Subsequently, they were homogenized in cold PBS and subjected to 10-fold serial dilutions until the desired concentrations were achieved. The dilutions were plated onto modified lauryl sulfate broth (mLST) plates and incubated at 37 °C for 24 h. After the incubation period, bacterial colonies were counted. The absolute quantities of *C. sakazakii* were determined based on their corresponding dilutions.

### 2.12. Serum Inflammatory Levels Analysis

The levels of inflammatory cytokines in the serum were analyzed via an ELISA assay kit (Nanjing Jiancheng, China). The inflammatory cytokines that were analyzed include IFN-γ, IL-6, IL-1β, IL-8, and TNF-α. The procedures were performed according to the instructions provided in the assay kit manual.

### 2.13. Gene Expression Analysis by qRT-PCR

RNA was extracted from tissues (n = 5) using an RNA extraction kit (ACCURATE, Changsha, China) according to the manufacturer’s instructions. Subsequently, the RNA was reverse-converted into complementary DNA (cDNA) via reverse transcription using a PrimeScript™ RT reagent Kit (ACCURATE, Changsha, China). Real-time quantitative PCR (qPCR) was performed in triplicate via the Mastercycler^®^ ep realplex system and TB Green™ Premix Ex Taq™ II (ACCURATE, Changsha, China). The specificity of the PCR products was analyzed by examining their melting curves. Relative gene expression levels were quantified via the 2^^(−ΔΔCt)^ method.

### 2.14. Histopathological Analysis

The histological samples were subjected to hematoxylin and eosin (HE) staining for analysis. A portion of the samples was fixed in 4% paraformaldehyde and subsequently underwent a series of processes, including dehydration, clearing, and embedding in paraffin, to obtain 5-μm sections. Furthermore, sections of the colon embedded in paraffin were stained using the hematoxylin and eosin (HE) staining methods.

### 2.15. Statistical Analysis

SPSS was used for data analysis. Statistical analysis was performed using Student’s unpaired *t*-test and one-way analysis of variance (ANOVA) to assess significant differences in the results. A significance level of 0.05 was used, and P values below this threshold were considered statistically insignificant. Data were expressed as mean ± deviation, and three independent replications were performed for each experiment.

## 3. Results

### 3.1. Construction and Validation of ΔESA_00986 and Complementary Strains

To investigate the molecular mechanism of *ESA_00986*, we generated an *ESA_00986* mutant strain via the homologous recombination technique. Figure 1A depicts the amplification sites targeted by the two primers (*ESA_00986* 1F/R, *ESA_00986* 2F/R) in the WT, *ΔESA_00986*, and cpESA_00986 strains. PCR verification was conducted using two primers with DNA from the WT, ΔESA_00986, and cpESA_00986 strains as templates. The results of the PCR analysis are presented in Figure 1B. When the ESA_00986 1F/R primers were used, both the WT and cpESA_00986 strains exhibited a visible DNA band of 1166 bp; the ΔESA_00986 strain exhibited no DNA band. However, when using the ESA_00986 2F/R primers, the WT strain showed a DNA band of 1620 bp, whereas the ΔESA_00986 and cpESA_00986 strains displayed a DNA band of 600 bp. The PCR results showed successful construction of both the ΔESA_00986 and cpESA_00986 strains, indicating their suitability for further experiments.

### 3.2. Effects of ESA_00986 Gene on Growth

In order to assess the impact of *ESA_00986* gene deletion on the growth rate and the OD_600_ of the WT, the *ΔESA_00986*, and *cpESA_00986* strains were measured via a UV-Vis spectrophotometer over a 14-h period. The growth rate results are presented in Figure 2. Comparing the growth rate of the *ΔESA_00986* with that of the WT strain, no significant difference was observed. Likewise, there was no significant change in the OD_600_ of the *cpESA_00986* strain compared to the WT strain. These findings suggest that the deletion of the *ESA_00986* gene does not impact bacterial growth. They also indicate that *ESA_00986* is not essential for cell growth, thereby eliminating potential variations in growth patterns as a confounding factor in future experiments.

### 3.3. Effects of ESA_00986 Gene on Cell Adhesion/Invasion

*ESA_00986* is predicted to be an Intimin/Invasin protein that is believed to play a role in bacterial adhesion to and invasion of host cells. The adhesion/invasion assay was performed to investigate the role of *ESA_00986* in the colonization of HCT-8 and HIEC-6 intestinal epithelial cell lines by *C. sakazakii*. Significant differences in colonization were observed between the WT, *ΔESA_00986*, and *cpESA_00986* strains in both cell lines (Figure 3). The relative *ΔESA_00986* strain adhesion/invasion rate of both HCT-8 and HIEC-6 cells was significantly lower compared to the WT strain. Furthermore, the *cpESA_00986* strain showed significantly higher adhesion/invasion compared to the *ΔESA_00986* strain in both cell lines. Similarly, the relative invasion rate of the three strains showed the same trend. These findings indicate that the *ESA_00986* gene plays a crucial role in the colonization of *C. sakazakii* in vitro.

### 3.4. Effects of ESA_00986 Gene on Virulence of C. sakazakii In Vivo

As a predicted invasin, *ESA_00986* has the potential to influence the pathogenicity of *C. sakazakii.* Three separate groups of 48 h old rats were orally infected with WT, *ΔESA_00986*, and *cpESA_00986* strains, and the survival of the animals was closely monitored for a period of 48 h following infection. Rats that were infected with the WT strain or *cpESA_00986* strains started experiencing mortality at 11 h, with 60% of the animals dying within 48 h in the group infected with the WT strain. In contrast, rats infected with the *ΔESA_00986* strain exhibited delayed mortality, starting at 17 h; 80% of the infected rats survived for at least 48 h. The survival curve (Figure 4) exhibited a significant distinction in pathogenicity between the *ΔESA_00986* and WT strains. Additionally, the colonization ability of *C. sakazakii* in rats was investigated in relation to the *ESA_00986* gene. The results revealed that the bacterial load of the *ΔESA_00986* strain was lower than that of the WT strain in the blood, brain, liver, and spleen (Figure 5).

### 3.5. Pathological Analysis

Histological analysis was conducted on brain and colon tissues collected from rats 48 h after bacterial infection. HE staining showed significant differences between the WT-infected group and normal group in brain tissue, including matrix dissolution and a loose, sponge-like structure (Figure 6A). In the *ΔESA_00986*-infected group, the brain matrix exhibited slight looseness and showed signs of inflammatory cell infiltration. The *cpESA_00986*-infected group had the same results as the WT group. After 48 h of infection, the intestinal tissue of rats infected with the WT strain was examined and signs of villus dilation, necrosis, perforation, and destruction were found in the intestine (Figure 6B). In contrast, the severity of intestinal tissue damage was notably reduced in rats infected with the *ΔESA_00986* strain. These findings suggest that knockout of the *ESA_00986* gene reduces the virulence of *C. sakazakii.*

### 3.6. Effect of ESA_00986 on the Serum Inflammatory Cytokines

Inflammatory factor is an important cytokine involved in immune regulation during the inflammatory response and inflammatory diseases. When the body is injured and undergoes a stress response, pro-inflammatory cytokines (such as TNF-α, IFN-γ, IL-6, IL-1β, etc.) are produced for immune regulation. At the end of the 48-h toxicity experiment, the blood of suckling rats was collected, and the serum of rat was obtained via centrifugation. The inflammatory factor in the serum was measured by detection kit. As depicted in Figure 7, the serum contents of pro-inflammatory cytokines IL-6, TNF-α, IFN-γ and IL-1ß were significantly reduced in the mutant group compared with the wild-type group. There was no significant difference between the complement group and the wild group. Thus, the decrease in proinflammatory cytokine levels implies a reduction in inflammation caused by bacterial infection.

### 3.7. Effect of ESA_00986 on the Expression of Genes Involved in Inflammation and Intestinal Integrity

The impact of *ESA_00986* on gene expression related to inflammation and intestinal integrity in the colon was assessed at the mRNA level, as depicted in Figure 8. After 48 h of infection, the *ΔESA_00986* group exhibited a significant downregulation in the expression of inflammatory factors, including IL-6, IL-1β, IFN-γ, and TNF-α compared with the WT group. Conversely, the expression of genes associated with gut barrier function, such as zonula occludens-2 (ZO-2), claudins-1 (CLA-1), claudins-2 (CLA-2), and occludins-1 (OC-1), were upregulated in the colon of the *ΔESA_00986* group. However, no significant difference was observed in the expression of zonula occludens-1 (ZO-1).

### 3.8. Effects of ESA_00986 Gene on the Motility of C. sakazakii

Motility, which is related to adhesion and invasion, was measured via swimming rings observed on the semi-solid medium. As is shown in Figure 9A, the size of the swim rings formed by WT, *ΔESA_00986*, and *cpESA_00986* on 0.3% agar medium has changed. Compared with the WT strain, the *ESA_00986* mutant displayed a loss of swimming mobility, whereas *cpESA_00986* showed a partial restoration in swimming mobility. As depicted in Figure 9B, the average migration radius of the WT strain was approximately 2.76 ± 0.25 cm, that of *ΔESA_00986* strain was approximately 1.73 ± 0.25 cm, and that of *cpESA_00986* was approximately 2.76 ± 0.30 cm. These findings provide evidence that the *ESA_00986* gene is crucial for the swimming mobility of *C. sakazakii.*

### 3.9. Effects of ESA_00986 Gene on Physiological Properties of Bacterial Surface

In order to assess the membrane permeability of the *ΔESA_00986* strain, bacteria were exposed to a hydrophobic fluorescent probe, 1-phenylnaphthylamine (NPN). An increase in NPN absorption was observed in the *ΔESA_00986* strain (Figure 10A). To compare the surface hydrophobicity of the WT strain and the *ΔESA_00986* strain, bacteria suspended in xylene mixture were trained for 3 h at room temperature; bacterial surface hydrophobicity was also reduced in the *ΔESA_00986* strain (Figure 10B). In summary, the absence of *ESA_00986* led to significant changes in the permeability and hydrophobicity of the bacterial surface.

### 3.10. Effects of ESA_00986 Gene on Biofilm Synthesis

In this study, we investigated the impact of the *ESA_00986* gene on biofilm formation in *C. sakazakii* ATCC BAA-894. Quantitative analysis was performed using crystal violet staining to assess the extent of biofilm formation. The WT strain and *ΔESA_00986* mutant strain showed significant differences in biofilm formation within 72 h. Compared with the WT strain, biofilm formation was significantly reduced in the *ΔESA_00986* mutant (Figure 10C). These findings indicate that the absence of the *ESA_00986* gene significantly impacts the formation of biofilms in *C. sakazakii* within 72 h. The *ESA_00986* gene has played a positive role in biofilm formation.

## 4. Discussion

Previously, we compared the adhesion and infection ability of 20 strains of *C. sakazakii* on HCT-8 cells, and found two strains with the strongest and weakest infection ability. Later, we conducted a comparative proteomic study on them and found an unknown and interesting protein, ESA_00986. The expression of this protein was much higher in strains with strong adhesion ability than in strains with low adhesion ability. This unknown protein (ESA_00986) from *C. sakazakii* BAA-894 showed homology with *Yersinia* and *Salmonella* intimin/invasin proteins. Therefore, we speculate that *ESA_00986* gene products may play a role in the pathogenesis of *C. sakazakii.*

In order to determine whether the *ESA_00986* gene is required for adherence to and invasion of host cells and for systemic infection, we examined the influence of the *ESA_00986* gene on the virulence of *C. sakazakii* in vitro using cell adhesion and invasion assays, and in vivo using a neonatal rat infection model. It was found that the mutant has defects in its ability to adhere to and infect intestinal epithelial cells, and the load of mutant strain in blood and tissues was significantly reduced. The ability of bacterial pathogens to adhere to and invade host cells, including macrophages and epithelial cells, plays a crucial role in bacterial pathogenesis. This ability is essential for systemic infection by and transmission of the pathogen. Before bacterial pathogens can successfully invade a host, they must first adhere to the surfaces of epithelial cells [36]. Invasins are proteins that allow bacteria to penetrate cells; they usually adhere to a cell before invading eukaryotic cells and also can help pathogens colonize, maintain, and spread pathogens within the host organism [37]. Deletion of the invasion gene *ychO* in avian pathogenic *E. coli* decreased its ability to adhere to and infect chicken fibroblasts [38]. In the present study, the results suggest that *ESA_00986* plays an important role in adhesion and infection by *C. sakazakii* and knocking out *ESA_00986* can reduce the toxicity of *C. sakazakii* in vivo, indicating that *ESA_00986* encodes an important virulence factor.

In this study, the animal experimental results showed that compared with the wild-type group, the levels of inflammatory factors (TNF-α, IFN-γ, IL-6, IL-1β) were reduced in serum and intestine of the mutant group. Inflammatory factors are an important class of cytokines that are involved in immune regulation during the occurrence of inflammatory response and inflammatory diseases. They are rapidly produced in large quantities under inflammatory conditions and are thought to be involved in a variety of inflammatory processes and induce a variety of inflammatory diseases [39,40]. IL-6, IL-1β, TNF-α, and IFN-γ are crucial regulators of NF-κB-mediated inflammation and play a critical role in the pathogenesis of colitis [41,42,43,44].

In comparison with the WT group, the expression of genes encoding ZO-2, occludin and claudin-1 in the intestinal tissue of the mutant group was upregulated, while their expression in the complementary strain group was consistent with the WT group. As demonstrated via HE staining of intestinal tissues, compared with the WT group, the mutant group also showed mild symptoms such as shedding of the villi of colon tissue, dilation of intestinal tissue, and thinning or destruction of the intestinal wall. The integrity of the intestinal epithelial barrier and the stability of the intestinal environment are widely recognized as crucial factors in maintaining intestinal health and promoting host resistance against pathogen invasion [45]. Tight junctions are integral components of the intestinal epithelial barrier and play an important role in preventing bacterial infection [46,47,48]. Tight-junction proteins ZO-1 and ZO-2 are primary proteins involved in the formation and maintenance of tight junctions. As markers of intestine mechanical barrier, they are closely associated with the integrity of epithelial cells. Occludin is also a crucial protein involved in maintaining the stability and function of tight junctions, contributing to the integrity of the barrier. Claudin-1 is a transmembrane protein that serves as a key component of tight junctions and plays a vital role in their structure and function. Tight junctions can prevent *Campylobacter jejuni* from invading laterally from epithelial cells. They play a major role in bacterial invasion. Pro-inflammatory cytokines, TNF-α, can induce tight junctions to break down and promote the invasion of *C. jejuni* [49,50]. Enteropathogenic *E. coli* can promote infection by breaking the tight connections of the small intestine’s epithelial cells [51]. A previous study showed that increasing expression of claudins and occludin can enhance intestinal barrier function of mouse and reduce *Salmonella* infection [52]. In agreement with these studies, the protein ESA_00986 can damage the intestinal barrier via inhibition the related genes expression (ZO-2, claudins and occludin) and enhance the intestinal infection ability of *C. sakazakii*.

In this study, it was found that compared with the wild-type, the biofilm forming ability of *ΔESA_00986* was reduced. Biofilms are commonly described as aggregates of microbial cells that adhere to both biological and abiotic surfaces, forming complex structures, consisting of a variety of major biomacromolecules that act as defensive barriers and important adhesion bases [53]. Numerous studies have demonstrated the important role of biofilms in the adhesion and invasion of human epithelial cells by pathogenic bacteria. Biofilms provides a protective environment that enhances bacterial attachment, colonization, and subsequent invasion of host cells, contributing to the pathogenicity of these bacteria [54]. The ability of bacteria to attach to surfaces and form biofilms contributes to successive infections [55]. Naziri et al. reported the role of *uropathogenic Escherichia coli* biofilms in the pathogenesis of initial attachment and invasion of the human urinary tract [56]. Hojjatolah Zamani et al. reported biofilm formation in *uropathogenic Escherichia coli* were association with adhesion factor genes, such as *papAH, bmaE* and *sfaS* [57]. *K. pneumoniae* biofilms is associated with colonization of the gastrointestinal, respiratory, and urinary tracts, as well as the development of invasive infections, particularly in immunocompromised patients [58]. *Cronobacter* has the ability to adhere to diverse surfaces and form biofilms, which enables it to withstand different stress conditions, enhance adhesion, and promote pathogenesis [59,60]. It is reported that *ompF* and *bcsR* of *C. sakazakii* can regulate adhesion/invasion by influencing biofilm synthesis [14,34]. As several members of the intimin/invasion protein family, compared with the wild-type, the biofilm forming ability of *ΔESA_00986* was also reduced. Consistent with these studies, the protein ESA_00986 may play a role in adhesion/invasion of *C. sakazakii* via regulating biofilm biosynthesis.

## 5. Conclusions

In this study, we demonstrated for the first time that deletion of *ESA_00986*, a gene encoding a virulence factor with Ig-like domains. The deletion of *ESA_00986* can lead to a notable reduction in invasion of epithelial cells and dissemination into the rat tissue. *ESA_00986* may played a positive role in adhesion/invasion through upregulation the intestinal inflammation via the NF-κB signaling pathway, damaging the intestinal barrier via inhibition the related genes expression and enhancing the biofilm synthesis. The ESA_00986 protein can be one of the best targets for future drug and vaccine development due to the interaction with the host immune system. This study provides valuable insights in the molecular mechanism and physiological function of the *ESA_00986* gene in *C. sakazakii.*

## Figures and Tables

**Figure 1 foods-12-02572-f001:**
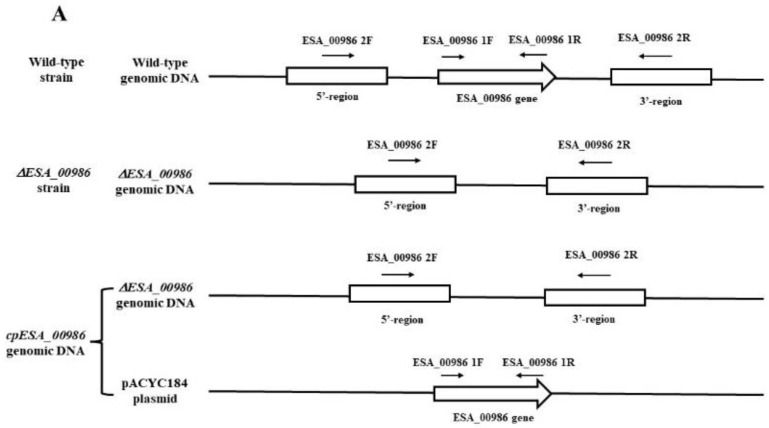
PCR validation of WT, ΔESA_00986, and cpESA_00986 strains (**A**) The positions of the two primer pairs of WT strain, ΔESA_00986 and cpESA_00986 strain. (**B**) PCR amplification results obtained via two primer pairs for the WT, ΔESA_00986, and cpESA_00986 strains. M, Marker; Lane 1, WT; Lane 2, ΔESA_00986; Lane 3, cpESA_00986.

**Figure 2 foods-12-02572-f002:**
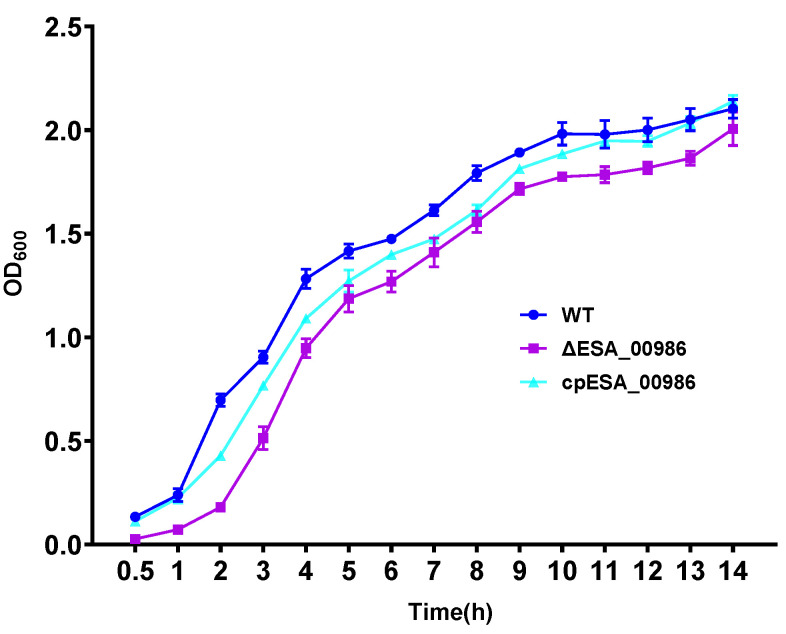
The growth curves of WT, *ΔESA_00986*, and *cpESA_00986* strains. Data are shown as mean ± S.D. Significant differences between *ΔESA_00986*/*cpESA_00986* groups and WT group were analyzed via one-way ANOVA. WT: wild-type strain; *ΔESA_00986*: mutant strain; *cpESA_00986*: complementary strain.

**Figure 3 foods-12-02572-f003:**
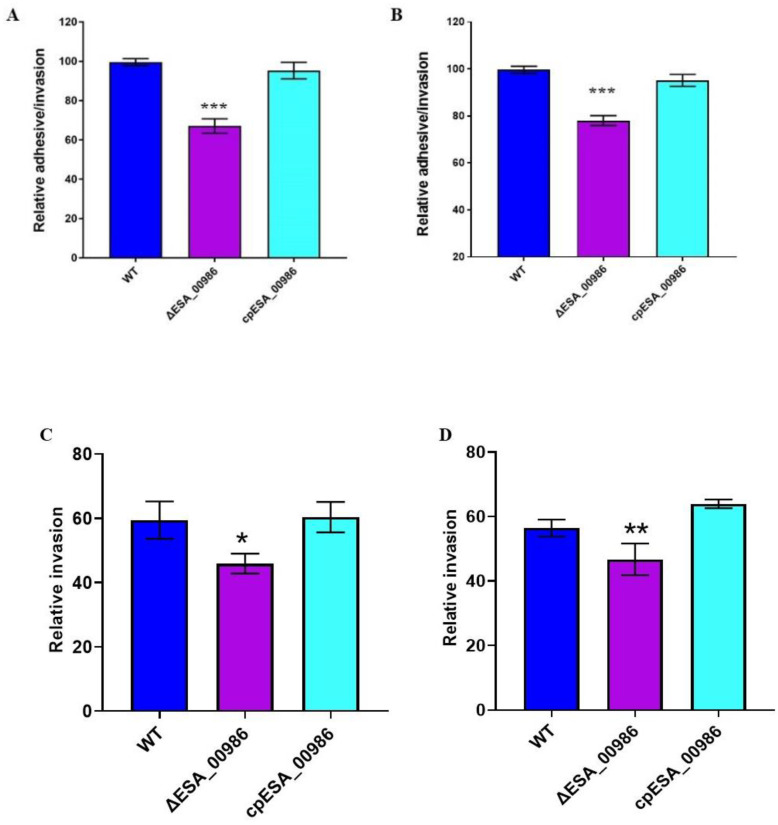
Effects of the *ESA_00986* gene on the adhesion and invasion. Data are shown as mean ± S.D. Significant differences between *ΔESA_00986*/*cpESA_00986* groups and WT group was analyzed at * *p* < 0.05, ** *p* < 0.01, and *** *p* < 0.001 via one-way ANOVA. WT: wild-type strain; *ΔESA_00986*: mutant strain; *cpESA_00986*: complementary strain. (**A**): the adhesion/invasion assay to the HCT-8 cells; (**B**): the adhesion/invasion assay to HIEC-6 cells; (**C**): the invasion assay to HCT-8 cells; (**D**): the invasion assay to HIEC-6 cells.

**Figure 4 foods-12-02572-f004:**
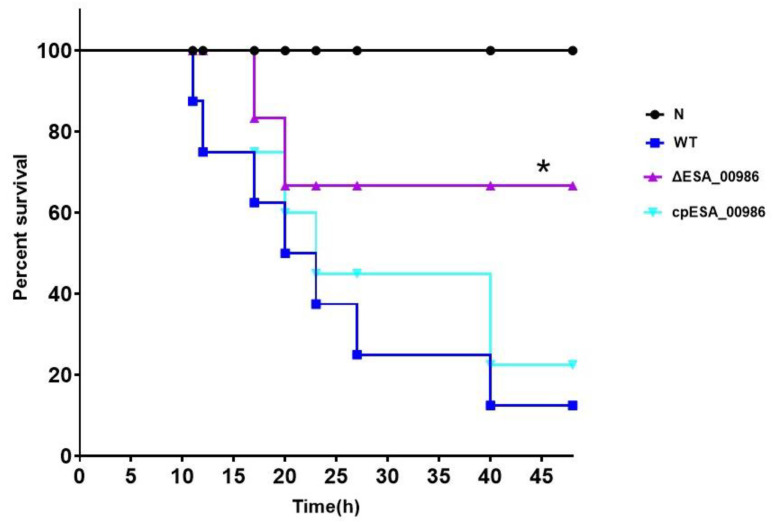
The survival curve of rats infected with *C. sakazakii* WT, *ΔESA_00986* and *cpESA_00986* strains. Significant differences between *ΔESA_00986*/*cpESA_00986* groups and WT group was analyzed at * *p* < 0.05 via one-way ANOVA. WT: wild-type strain; *ΔESA_00986*: mutant strain; *cpESA_00986*: complementary strain.

**Figure 5 foods-12-02572-f005:**
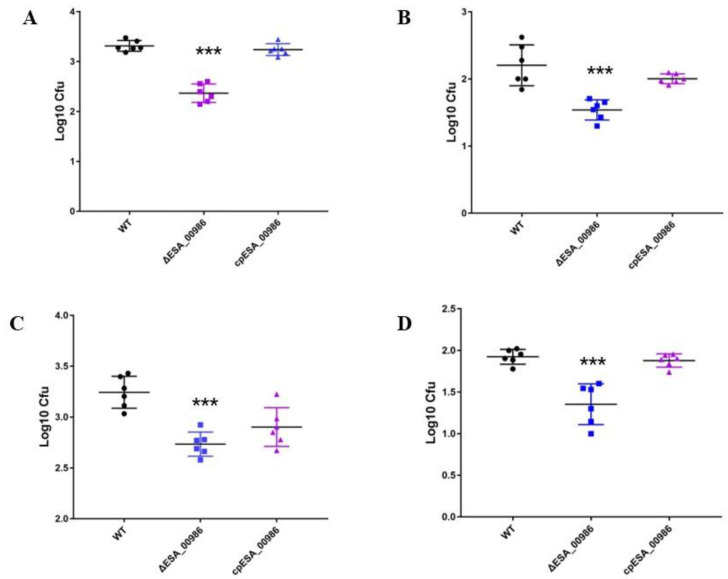
The bacterial load of *C. sakazakii* WT, *ΔESA_00986* and *cpESA_00986* strains in blood, brain, liver, and spleen. Data are shown as mean ± S.D. Significant differences between *ΔESA_00986*/*cpESA_00986* groups and WT group was analyzed at *** *p* < 0.001 via one-way ANOVA. WT: wild-type strain; *ΔESA_00986*: mutant strain; *cpESA_00986*: complementary strain. (A): Brain; (B): Spleen; (C): Blood; (D): Liver.

**Figure 6 foods-12-02572-f006:**
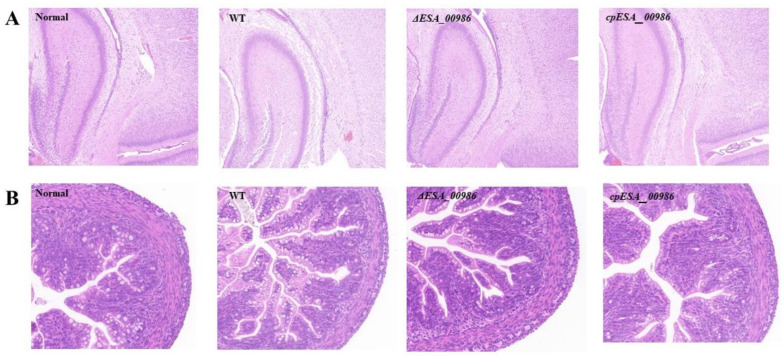
The brain tissue section (**A**) and the colon section (**B**) of *C. sakazakii* WT, *ΔESA_00986,* and *cpESA_00986* strains. WT: wild-type strain-infected group; *ΔESA_00986*: mutant strain-infected group; *cpESA_00986*: complementary strain-infected group; NC: normal group.

**Figure 7 foods-12-02572-f007:**
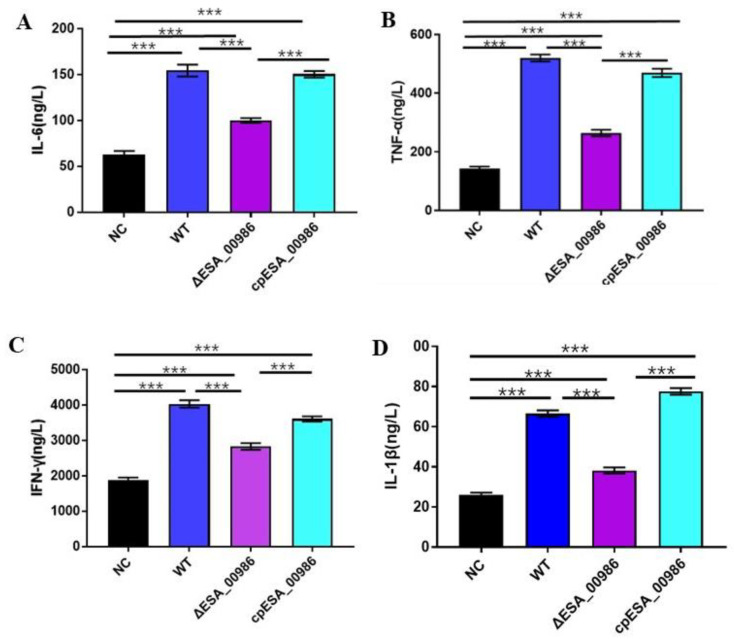
Inflammatory factor content in serum ((**A**): IL-6, (**B**): TNF-α, (**C**): IFN-γ, (**D**): IL-1ß)). Data are shown as mean ± S.D. Significant differences between *ΔESA_00986*/*cpESA_00986* groups and WT group were analyzed at *** *p* < 0.001 via one-way ANOVA. WT: wild-type strain-infected group; *ΔESA_00986*: mutant strain-infected group; *cpESA_00986:* complementary strain-infected group; NC: normal group.

**Figure 8 foods-12-02572-f008:**
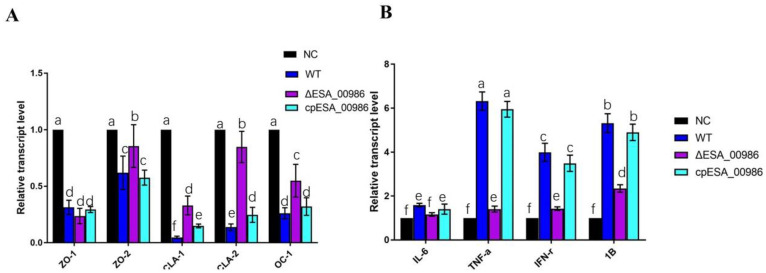
Gene expressions of tight-junction proteins (**A**) and inflammatory factors (**B**). Significant differences are indicated by different letters (a–f) (*p* < 0.05). WT: wild-type strain-infected group; *ΔESA_00986*: mutant strain-infected group; *cpESA_00986:* complementary strains infected group; NC: normal group.

**Figure 9 foods-12-02572-f009:**
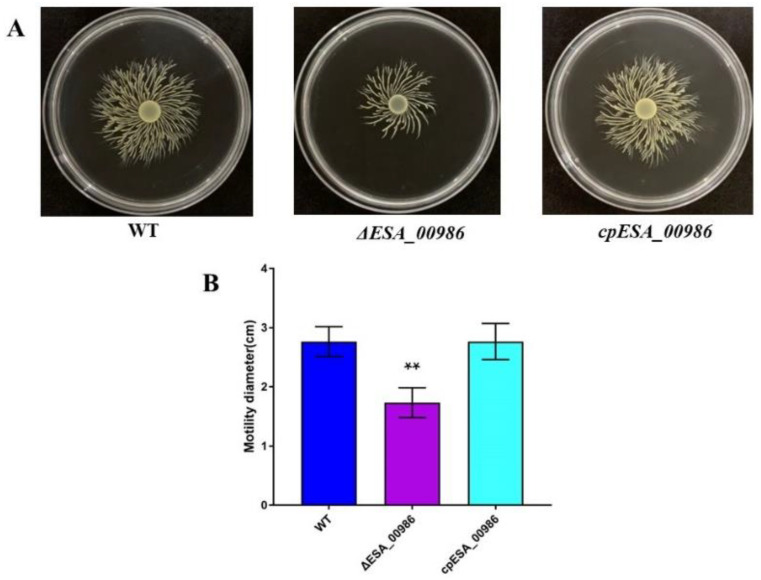
Mobility of the WT, *ΔESA_00986* and *cpESA_00986* strains on semisolid medium. Data are shown as mean ± S.D. Significant differences between *ΔESA_00986*/*cpESA_00986* groups and WT group were analyzed at ** *p* < 0.01 via one-way ANOVA. (**A**) The white translucent rings observed represent the migration rings of the bacteria. (**B**) The respective migration radii of the WT, *ΔESA_00986*, and *cpESA_00986* strains.

**Figure 10 foods-12-02572-f010:**
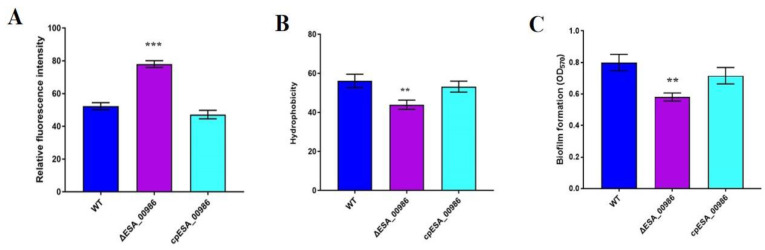
The membrane permeability (**A**), surface hydrophobicity (**B**) and biofilm formation (**C**) of *C. sakazakii* WT, *ΔESA_00986* and *cpESA_00986* strains, respectively. Data are shown as mean ± S.D. Significant differences between *ΔESA_00986*/*cpESA_00986* groups and WT group were analyzed at ** *p* < 0.01, *** *p* < 0.001 via one-way ANOVA. WT: wild-type strain; *ΔESA_00986*: mutant strain; *cpESA_00986*: co mplementary strain.

**Table 1 foods-12-02572-t001:** Bacterial strains and plasmids used in this study.

Strain or Plasmid	Genotype or Characteristics	Source
*Cronobacter sakazakii*		
ATCC BAA-894	WT	ATCC
*Δ* *ESA_00986*	*ΔESA_00986*::amp^r^	This study
*cp* *ESA_00986*	*ΔESA_00986* with pACYC184-*ESA_00986*	This study
*E. coli*		
S17 lambda pir	Strain for construction harboring lambda pir	29
Plasmids		
pCVD442	Suicide plasmid for deletion: amp^r^	29
pCVD442-Q-H	pCVD442 with homologous arms	This study
pACYC184	p15A ori Cm^r^ Tet^r^	
pACYC184-*ESA_00986*	pACYC184 with *ESA_00986*	This study

**Table 2 foods-12-02572-t002:** Primers used in this study.

Primer	Gene Amplified	Primer Sequences (5′-3′)	Amplification Size (bp)
pCVD442 F	Suicide plasmid for markerless deletion	CAATAACCCTGATAAATGCTTCAA	6345
pCVD442 R		CTCATGAGCGGATACATATTTG	
*ESA_00986* QF	Upstream of *ESA_00986* gene	TGATAAATGCTTCAACGTCAGCGTCACCTGGAACG	828
*ESA_00986* QR		GTGGTAGTTCAGGCGACTATTTTCCTGACGGAAACAGACG	
*ESA_00986* HF	Downstream of *ESA_00986* gene	ATAGTCGCCTGAACTACCACAAATGG	780
*ESA_00986* HR		GCGGATACATATTTGCCAGCCCGTCAGTCGTAATG	
*ESA_00986* 1F	*ESA_00986* gene sequence	TACATGGCAACTTATTCTGTATTTAC	1166
*ESA_00986* 1R		TGACCATTTGTGGTAGTTCAGG	
*ESA_00986* 2F	Both ends of the *ESA_00986* gene sequence	GCTCTGTCTCGGTGATGTAG	1620(WT)/600(*ΔESA_00986*)
*ESA_00986* 2R		TGATCCTTCAATCCCAGTAA	

## Data Availability

The data presented in this study are available on request from the corresponding author.

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
