# Peer review of "Effects of ESA_00986 Gene on Adhesion/Invasion and Virulence of Cronobacter sakazakii and Its Molecular Mechanism"

_foods, 2023, doi:10.3390/foods12132572_

Round 1

Reviewer 1 Report

I must appreciate the authors elaborative work on the targeted  gene ESA_00986  of C.sakazakii which is one of the emerging human pathogen especially for infants 

The work is well designed and presented but is too lengthy. The introduction can be reduced keeping the relevant information only and the repetition of discussion can be avoided without losing the essence of the  article. 

The prevalence and actual burden of C.sakazakii can be included in the introduction in short.

In the title the "Sakazakii" is species so "sakazakii" should be used, please  correct. Check italics for the C.sakazakii in various places like line 56.

Jerse et al year should be mentioned and to be included in reference

Future prospects of the studied protein can be included in discussion/conclusion

Reviewer 2 Report

The manuscript: Effects of ESA_00986 gene on adhesion/invasion and virulence of Cronobacter Sakazakii and its molecular mechanism from Fan et al is a very interesting manuscript. However, it needs a revision on the style and on the written english. Moreover, there are several sentences with missing references. 

Lines 85-90: These sentences do not correspond to the introduction section because they explain the methodology and indicate a conclusion, so they should be deleted.

The discussion is a little bit jumbled. The normal structure for the discussion is to present the results first and then to discuss about them. 

For instance, it seems that from lines 377 to 389 the authors present results from previous studies and do not include references.

Other comments:

Line 73: Jerse et al., a reference is missing.

Line 80: In our previous study, a reference is missing

Lines 106-107: Commercially available reagents were utilized for this purpose… Which reagents?

Line 126: photometer.. Which instrument? Photometer or spectrophotometer?

Line 136-137: previously established methods, a reference is missing.

Line 245: UV-Vis spectrophotometer…Is this the same instrument mentioned before?

Lines 377-389: References are missing, not only for “previous results” but also for other explanations included.

388-389: “Therefore, we speculate that ESA_00986 gene product may play a role in the pathogenesis of C. sakazakii.” Is this speculation from this study? Or is it from previous studies?

390-396: No references are included.

English should be carefully revised. Here you have some examples of sentences that should be improved or re written.

Lines 37-38: It is a facultative anaerobes bacteria. Correct this sentence (It is or They are?)

Line 56: C. sakazakii in italics

Line 73: Escherichia coli should be E. coli and it should be written in italics

Line 149: 600 nm (OD600) This is redundant. The authors should use the same abbreviation all along the manuscript. In other sentences they use subscript. The abbreviature should be used the first time that it is written in the manuscript.

Line 198: kit(Nanjing Jiancheng,China). A space is missing here and in several other sentences.

404: in vivo should be in italics

432: C. jejuni

Enteropathogenic, uropathogenic or enteroaggregative should be written in normal letters, not in italics.

436: E. coli. Moreover, a space is missing in [50].Previous

Reviewer 3 Report

The manuscript by Fan and authors aims to determine the role of ESA_00986 in the pathogenesis of Cronobacter sakazakii.  The authors constructed a defined mutant in the gene using a suicide plasmid and constructed a plasmid complement of the gene in pACYC184.  The authors compared growth effects, biofilm formation, surface characteristics, motility, adhesion/invasion and pathogenicity in a rat model.  This study is interesting but the manuscript requires significant changes.

The authors described the gene of interest, ECA_00986 as a putative invasin/intimin identified from a previous study.  The authors did not provide the reference for the study where this putative virulence factor was identified by proteomics and described supporting sequence alignments.  This information is required for the readers to understand the similarities of this putative virulence factor with other characterised orthologues in other pathogenic bacterial species.  The authors have at least three previous studies on Cronobacter species which, from a quick look, do not include the comparisons that they describe for ECA_00986 (https://doi.org/10.3389/fmicb.2020.01239, DOI 10.1186/s13568-016-0246-4)

The study describes ECA_00986 as a putative invasin but the in vitro assays do not test its role in the bacteria's ability to invade epithelial cells, by including gentamycin treatment to quantify intracellular bacteria in the experimental design; this data with the attachment data would strengthen the molecular characterisation for its role in the pathogenicity of Cronobacter.

In the Introduction and Discussion, more emphasis could be included on why understanding the mechanisms of interactions of the bacteria are needed for a food safety perspective for the scope of the journal.  Can Cronobacter adhere to milk proteins?

For the motility assay, the bacteria were grown at 37 degrees for the overnight culture, but the assay performed at 30 - is there a biological reason for this?  

Figure legends do not describe which statistical test was applied to the data.  Many of the figures appear pixelated and some axis labels have very small font sizes, making it difficult to read.

Other specific comments:

Line 16 – other corresponding author not in list of authors under title

Line 24: Invasitin for invasin

Line 61 - ‘LysR regulator critical role in adhesion and invasion of intestinal cells’ : needs rephrased – LysR regulator critical role in the regulation of factors involved in adhesion and invasion of human intestinal cells.

Line 65 – Statement needs rephrased as not all adhesins stimulate cytoskeletal rearrangements; e.g. adhesins are present on fimbriae for initial attachment to surfaces.

Line 118 – 00986 gene fragment – what part of the coding sequence was cloned?  Was any UTR included in the complement?

Line 312 – mentions ‘lactating rats’ – why?

Line 384-384 – repeat of statement intimin/invasion function

Line 465 – ECA_00986 does not regulate but plays a role in adhesion/invasion

Line 468 – italicised ‘can’

There needs to be review on the use of grammar and there are occasions where statements are repeated; e.g. Line 44-45 and Lines 384-396.
